# *Get Back*: The New Galician Diaspora Goes on Stage

## María Alonso Alonso

Department of English and German Philology, University of Santiago de Compostela,
15782 Santiago de Compostela, Spain; maria.alonso.alonso@usc.es

**Abstract:** This article analyses *Get Back* (2016), a play written by Diego Ameixeiras and directed by Jorge Coira. The text will be considered an example of an early Brexit narrative, and it will serve to explore how the new Galician diaspora is represented through the arts. Issues related to migration, racism, and precariousness bloom naturally from a play that gathers four Galician migrants in London, together with a British-born character, inside one of the carriages of the Tube. Old and new waves of Galician migrants will be juxtaposed through different characters, illustrating the complexity of this recent migratory phenomenon. Several stereotypes will be exposed to increase how Ameixeiras constructs generational and gender gaps existing among Pepe, Luisa, Rafa and Iria, four immigrants who find themselves sharing a carriage on the London Underground sometime during the aftermath of Brexit. Thanks to the multiple dichotomies and arguments that create an ambivalent sense of Galician identity abroad, the play runs very smoothly. The different points of view found in the text will reflect on the subaltern status of the characters, who seem to struggle to find their place in their host country.

**Keywords:** new Galician diaspora; emigration; immigration; gender; precariousness; diaspora space; subalternity; racism; Brexit

## 1. Introduction

Of all the Tube trains that travel across the London Underground every day, what are the chances that four Galician migrants find themselves sharing the same carriage? This coincidence is the departing point of *Get Back*, a play written by Diego Ameixeiras (2016, unpublished) and directed by Jorge Coira. Iria, Rafa, Pepe and Luisa are stuck for a few hours inside the same carriage for security reasons after an unknown person commits suicide on one of the lines on their route. This incident allows the protagonists of this play to reflect on their migrant status in an increasingly hostile country. During the hours that these four Galician characters are forced to share, together with British-born citizen John, they all talk about migration, precariousness, alienation, uprooting and the idea of returning to the homeland. The play takes the name of one of the most famous songs released by The Beatles in 1969; it dealt with issues related to the United Kingdom's immigration policies. Eme2[1] is the company that put this play on the stage on the 23 September 2016, three months after the Brexit referendum took place in the United Kingdom with the well-known outcome.

The idea for the play is based on the research work undertaken by journalist Camilo Franco, who in 2015 interviewed a cross-section of Galicians living in London who belonged to different generations and areas of employment. These interviews inspired Ameixeiras to write a text to represent "a realidade fraccionada da Galicia contemporánea" (González 2016). *Get Back* contributes to the debate on representing the new Galician diaspora within the arts by demystifying this diasporic phenomenon through various comic devices. It is possibly because of the ambivalent nature of this play that Coira defines it as a 'comedy-drama' (Castro 2016). The text, indeed, raises issues related to migration, and it does so by challenging the stereotypical image of the new Galician migrant promoted

by the media through TV programmes such as "Galegos polo mundo" (TVG) or "Calle-jeros viajeros" (Cuatro). *Get Back*, thus, turns into a counter-discourse that questions the over-optimistic view offered on the media around the idea resulting from leaving one's home country in search of better opportunities. As the following analysis will illustrate, the new Galician migrant that Ameixeiras represents in his text is highly critical of the social and political circumstances that took them abroad, a view that, allegedly, differs from the previous generation of migrants in the United Kingdom.

There is a dichotomy between Pepe and Luisa, on the one hand, and Rafa and Iria, on the other.[2] Pepe and Luisa belong to the first wave of Galician migration that arrived in the United Kingdom during the second half of the twentieth century and went through different vital experiences in the host country. On the other hand, Rafa and Iria belong to the new Galician diaspora and are still looking for their place in the capital city. A clear generational gap between these two groups of characters is illustrated through the motivations and results of their migrant experiences. Although these protagonists are built around many stereotypes, their characterisation represents the complexity of a new migrant phenomenon. This phenomenon is now becoming more present in Galician literature thanks to the works of, among others, Anna R. Figueiredo (2018), Eva Moreda (2020) and María Alonso Alonso (2021).[3]

The following pages will focus on analysing *Get Back* as an example of an early Brexit narrative. Accordingly, the London Tube will be considered an instance of a diaspora space, a "space in-between" (Bhabha 1995) that serves as a catalyst for a particular hybrid social and political consciousness to appear that favours the introspection needed to talk about racism, subalternity and the United Kingdom as a failed multicultural nation. Moreover, the migrant mantra of 'work and sacrifice' will articulate the analysis of the different protagonists according to their attitude towards their present and immediate future in the host country. Through these characters' hopes, expectations, and frustrations, *Get Back* manifests itself as an accurate depiction of how the new Galician diaspora challenges outdated paradigms around classic Galician migration.

## 2. Brown Line, Bakerloo, towards Elephant and Castle

It is not midday yet when the train that is taking our protagonists to their destinations stops abruptly. They remain in the darkness for a few seconds while checking their mobile phones to find out what is going on. What seems to be a simple technical fault holding up their journey for just a couple of minutes turns into a major incident that will keep them in that carriage for several hours. Iria is on her way to the airport to pick up her boyfriend, who had just arrived in London, intending to stay in the capital to find a job and start living with her girlfriend after four years of relationship. She is already running late, as she communicates through the messages she sends to him over the phone. Rafa is also late. He is on his way to a vital interview that might allow him to work in an IT position for an important company. Luisa is drunk, as usual, and she even finds the situation quite funny. Pepe notices her state and keeps staring at her with a recriminatory look. John continues reading a book when the light comes back on. Rafa is the first one who talks aloud to ask if someone has any further information about the incident. He is clearly distressed, moving from one place to another and shouting while trying to smash the doors open. He seems to be suffering from claustrophobia and faints, resulting in the other passengers approaching him to offer help. Once he recovers, Pepe asks him where he is from and when they realise that they are all Galician apart from John,[4] the play moves from English to Galician. They cannot imagine that they will end up being stuck in that carriage together for several hours at this early stage of the play.

Despite the confusion of the first few minutes and the defensive attitude they all show during most of the play, the London Tube serves as the ideal locus to reflect on the differences and similarities of old and new Galician diasporas. The carriage turns into a kind of limbo, a parenthesis in their lives that allows them to think and talk about how they relate to their home and host countries. Although they are stuck in the same location,

issues related to movement and dislocation bloom naturally from their conversations. The fact that there is a native English citizen with the other four Galician characters turns the carriage into a contact zone; that is, the carriage becomes what Avtar Brah (1996), in her foundational text *Cartographies of Diaspora: Contesting Identities*, would refer to as a 'diaspora space'. For Brah, the concept of 'diaspora space' not only refers to the conceptual category that migrants inhabit together with the natives of a given host country, but also to the post-colonial sense of place of our times as the result of different imperialist processes. That is, it is a concept that varies from one place to another and from one historical period to the next. In *Get Back*, the carriage is indeed a place where those who arrived from outside the national borders interact with those nationals who are affected in some way by their migration, but it also represents the ambivalent relationship nationals and non-nationals, as well as between old and new generations of migrants. It is a place where the different characters are forced to negotiate alliances regarding their in-between status, building on an abstract interpretation of space. The London Tube is instrumental in this respect. It is a metaphorical melting pot where people from different countries and statuses coincide for a few minutes every day. In this situation, the London Tube forces and encourages contact between people, who, despite having so much in common, would never communicate if not for the incident that motivates the play. Of course, they do not know this at the beginning. Sharing a common homeland and a common language might seem reason enough to feel close to each other, but this is not the case here. On the contrary, the generational and gender gaps between the four Galician characters increase their alienation from their host country and each other.

Galicia is often thought of with respect to its history of migrations and characterised by its intra and transnational flows. José Colmeiro (2018) considers that Galician diasporas, both in the past and present, have contributed to a distinctive sense of glocal identity, understanding the term 'glocal' as one that stands for the universal and particular peculiarities of contemporary social, political, and economic systems. The Galician diaspora has turned into what Stephen Vertovec would refer to as a 'transnational community', "sustained by a range of modes of social organization, mobility and communication" (Vertovec 2009, p. 3). For Vertovec, the term 'transnationalism' stands for "a condition in which, despite great distances and notwithstanding the presence of international borders [ . . . ], certain kind of relationships have been globally intensified" (p. 3). As a matter of fact, transnationalism has become an important characteristic of migrant experiences nowadays, through which old and new diasporic movements navigate depending on a number of factors. In this particular case, the truth is that the Galician migrant community has allowed the flow of Galician culture and identity beyond its borders. However, Galician migrant experience has changed radically in the last decades. High-speed travel, digital technologies and a different historical context characterise the new Galician diaspora.[5] This characterisation is evident from the beginning of the play when Iria is texting her boyfriend, who is waiting for her at the airport. Instantaneous communication allows Iria to maintain a permanent link with her people back home, although her boyfriend had just arrived in London. This communication implies that the new Galician diaspora needs to relate to the concepts of 'time' and 'space' differently from previous diasporas. However, immobility, as in this case, does not necessarily imply keeping communication on hold.

There is a clear difference in how old and new diasporas relate to Galicia as their home country in the play. As migrant subjects, all the Galician characters experience 'dislocation' (Ashcroft et al. [1989] 2004) in a particular way, which implies the deconstruction of their Galician identity and the subsequent reconstruction of a new sense of self. This reconstruction is built upon the different causes that motivate their transterritorialisation. As Coira himself acknowledges, for the new Galician diaspora "[a] morriña ten moito menos peso porque non é difícil volver pero ao mesmo tempo non acabas de estar nin nun sitio nin noutro" (Castro 2016).[6] Pepe, as a member of the classic Galician diaspora, is the one who experiences a sense of sentimental uprooting when he thinks about his homeland: "A avoa Josefa. Facía uns chourizos que sabían a gloria [ . . . ] Nunca comín chourizos coma

aqueles. Nin coñecín mellor persoa que a avoa Josefa" (p. 60). Pepe's childhood memories always take him to Galicia; to an imaginary Galicia that does not necessarily correspond with the 'real' Galicia that he left. As Arjun Appadurai (2003) refers to when theorising diaspora, there is a tendency to re-create an imaginary memory of the absent place by migrant communities. Appadurai identifies this need to create an idealized image of the homeland with what he calls "the imagination as a social practice" (p. 33), that is, a cultural process that shapes collective memory. For Pepe, Galicia is that imaginary homeland that he left when he was a child; it is a primitive, ahistorical, out-of-time location that only exists in his memory. Pepe is the son of two Galician migrants in Lausanne; that is, he is a migrant son of migrants. Although Galicia is the location of his better memories, it could be argued that his 'morriña' is more connected to some particular people rather than to a specific place. Although Galicia did not give him or his parents the opportunities or stability they were looking for, he does not manifest any kind of scorn. For Pepe, migrating is as natural as life itself. Something completely different happens with Iria, who insists on blaming her home country for expelling her: "Tiven un traballo de merda en Galicia, e cando acabou o contrato, marchei para Barcelona" (p. 25), "todo é unha merda" (p. 26). Iria feels disappointed with the outcome of her first steps into adulthood. She belongs to a generation of degree-holders who could not develop their career as expected and blame both their home and host countries for this.[7]

Despite apparent differences between new and old diasporas, migration still entails "both material and psychological gain as well as loss" (Alonso Alonso and Barbour 2020, p. 398). Luisa is possibly the character who best represents the idea of loss in the play. She is in her fifties and embodies the concept of failure: failure both as a migrant since she has not made a fortune and a failure as a mother through losing custody of a child whom she could not look after and so had to be sent back to Galicia to live with his grandparents. Subsequently, she completely lost contact with her mother, but this was not why she turned into an alcoholic. Luisa exemplifies what Forcadela (2013) refers to as a 'castration' when talking about Galician migration. In her particular place, this castration is both geographical and maternal; it is physical and psychological at the same time. "Deixáronme sen nada" (p. 55); she regrets when she decides to share her personal story at the end of the play. "Volvinme tola. Durmín na rúa. Nos parques. Espertei en sitios onde nunca debería espertar ninguén. A niña nai e o meu irmán viñeron para axudarme, pero non me deixei. Non quería axuda" (p. 56). Luisa represents an example of the 'bad' immigrant who does not contribute with her work and effort to enriching the country that hosts her. She is the antagonist of Pepe, the 'good' immigrant who succeeded and is grateful to the United Kingdom for giving him the opportunity to have his own business and who does not allow Iria to openly express her views on the precariousness of her situation in the United Kingdom "[p] orque en vez de agradecer que tes un traballo, non paras de queixarte" (p. 27). Pepe is the antithesis of both Luisa and Iria. Despite his annoying attitude throughout the play, he is possibly the character who experiences the most radical evolution as hours pass by, as mentioned at the end of this article.

## 3. Do Not Bite the Hand That Feeds You

"Pois se cadra debiches estudar algo de máis proveito, en vez de cuspir na man de quen che está axudando" (p. 28), Pepe recriminates to Iria when she complains about her precarious situation in London. Pepe's overindulging view of the United Kingdom as a host country causes most of his arguments with the two female characters in the play. As previously mentioned, Iria is not happy with her job conditions in London, multi-tasking at a Mexican restaurant as a waitress, kitchen porter, housekeeper, and kitchen assistant. She embodies disappointment and frustration, although she is possibly the biggest realist of all the characters. As she explains, she is a "bióloga. Catro anos de carreira con grao e posgrao, e non o fixen para acabar limpando merda" (p. 24). She was the student at her high school who attained the highest grade back in Galicia. She followed the path expected of her: to university to finish a degree with one of the higher marks of her class,

and, after that, she only found precariousness.[8] She is the counterpoint to the idealised image of contemporary migration, necessarily linked to success in the media. She also represents what Boswell and Geddes (2001) would refer to as the 'from brain drain to brain waste' dynamics that characterise this not-so-optimistic migrant experience. Galicia, in this case, spent thousands of euros in educating people such as Iria and this investment never returns. What is more, as Iria exemplifies, this investment is not even of any use in her host country since she does not find an opportunity to work in a job according to her education and skills.

Even though Iria and Rafa belong to the same generation of migrants and live in a precarious situation in their host country, their attitude towards their future is different. Iria lost all her faith in finding a better job in London as she acknowledges that when she is printing her CVs to apply for some jobs, "xa sei que non van valer absolutamente para nada" (p. 26). Rafa, on the contrary, keeps a positive attitude. He is actually on his way to an interview that means he has the possibility of securing the opportunity that he had been seeking since he arrived in London. He is the kind of migrant who believes that "[a]quí as cousas funcionan. Aquí valórase de verdade o esforzo. E se un traballa e pelexa polo que quere, acaba saíndo adiante" (p. 31). Unfortunately, his hopes and expectations will not materialise when he misses his interview; after going through a ridiculously long process of four different stages to obtain a simple job as an IT guy. When he finds out that the job was offered to someone else, he feels closer to Iria, and he considers that "nunca na puta vida vou conseguir un traballo" (p. 58). Reality clashes with his expectations. It is because of their precariousness that the idea or return is a constant in the play. Rafa, despite his fake optimism, even considers this possibility since his mother seems to keep insisting on him going back to Galicia: "Xa non teño vinte anos. Á mina idade, o meu pai, que en paz descanse, xa estaba farto de traballar e criara tres fillos" (p. 59). Through their hope and disappointment, Iria and Rafa illustrate a sense of generational failure of those who "believed their efforts would be rewarded and that has now been forced to come to terms with the fact that this may not happen, at least not for the foreseeable future" (Alonso Alonso 2018, p. 34).

Rafa and Iria (and Luisa, by extension) are the subaltern subjects through which the new Galician diaspora, the frustrated and unsuccessful one, speaks. In her celebrated essay *Can the Subaltern Speak?*, Gayatri C. Spivak (1988) coins the term 'subalternity' to refer to how discourses shape both individuals and communities. She draws on postmodernist theories to reflect on issues of agency and representation to depict the experience of those dispossessed from certain rights and presumably silenced by an empowered elite. In this respect, there is a clear difference between the depiction of the new Galician diaspora in the media and through the arts. Whereas the media insists on showing a positive and optimistic view of migration (Alonso Alonso 2017, p. 41), creative practices such as literature or the visual arts adopt a more critical attitude towards this new migratory phenomenon. They achieve this by focusing on the depreciation of the migrant subaltern community as human capital, which is how it is depicted in the play.[9] Brexit increases the alienation of subaltern subjects since their migrant status has become increasingly difficult. *Get Back* is, after all, extremely critical towards the consequences of the outcome of the 2016 referendum.

One of the arguments between Iria and Pepe about the United Kingdom as being a land of opportunities took place to discuss Brexit. As the diasporic limbo that it represents, the London Tube allows them to reflect on how the aftermath of the referendum, which took place three months before the time of the narration, will affect them. Despite the lack of historical distance, all of the characters, apart from John, recognise that they will become collateral victims of Brexit. It is at this point when John, the British-born character of the play, tells Iria that he thinks the United Kingdom is "um país moito hospitaleiro", to which Iria responds sarcastically: "Que o dis? Polo Brexit?" and continues giving examples of how racist attacks were already increasing all over the country, as when "[a]o cociñeiro do restaurant no que curro, que é colombiano, rompéronlle os cristais da casa" (p. 29). Thus, despite the previously mentioned lack of historical distance, *Get Back* succeeds in

portraying a dark future for European migrants in the United Kingdom, above all for those who live in total precariousness.

Apart from migration and precariousness, racism is one of the main topics of the play. *Get Back* differentiates between what John McLeod would refer to as 'external vs. internal racism'. He defines 'external racism' as "a form of xenophobia when groups of people who are located outside the nation's borders are discriminated against by the groups of their 'race'". In contrast, 'internal racism' is "directed at those who live within the nation but are not deemed to belong to the imagined community of the national people due to their perceived 'race'" (McLeod 2010, p. 133). Pepe is a paradigmatic example of the ambivalence of these two concepts. As Luisa affirms, he is "inmigrante e es un cabrón dos pés á cabeza" (p. 51). For him, who lived in the United Kingdom long enough, "Inglaterra xa non é o que era" (p. 30) and that is due to immigration, despite him also being an immigrant. He seems to agree with Brexit because he considers that there are too many immigrants in the United Kingdom who are not contributing, as he does, to the improvement of the nation. At the same time, he is highly critical of the immigrants that work legally in the country since, as the owner of a cleaning company, he keeps struggling against what he refers to as lazy migrants. Despite this, he considers the United Kingdom could not afford to expel all immigrants because that would not be "un bo negocio" (p. 29). Pepe is a racist. His immigrant status does not moderate his xenophobia to the point that he smiles out of pleasure when he is told that the man who committed suicide by throwing himself onto the tube line was a Senegalese illegal immigrant who was escaping from the police after stealing a mobile phone. Iria, as his antagonistic character, adopts the direct opposite stance on migration, whether it is legal as in her case or illegal as the Senegalese. For Iria, Brexit is the expected result of a racist country such as the United Kingdom, a place "cheo de racistas que pensan que lles estamos quitando o traballo" (p. 30).

As a result of Pepe's arguments throughout the play, he appears to embody an example of a 'mimic-man' (Bhabha 1995). A 'mimic-man' is a subaltern subject who adopts and perpetuates the attitude of the oppressor but, instead of moving upwards on an imaginary social scale, he remains in the same subaltern position. He is as racist as those who voted for Brexit despite being one of the reasons that motivated Brexit. He is an immigrant against immigration, an ambivalent character that represents the idiocy of xenophobia in the play. Despite this and his defensive attitude throughout the play, Pepe can reconcile with his immigrant nature and his compatriots. By the end, he apologises to Luisa and convinces her to fly with him straight back to Galicia to reunite with her son. They leave the other three passengers in the carriage shocked with his change in his attitude and his solidarity towards a compatriot who is struggling to be afloat.

### 4. Conclusions

*Get Back* finishes with hope when Pepe and Luisa abandon the carriage after transit is resumed. He offers her to pay for her trip back to Galicia and accompany her to reencounter with her son. He wants to help someone he has nothing in common with and without asking for anything in return. They are the first ones to leave the carriage. Still in shock by Pepe's change in attitude, Iria, Rafa and John take a picture as a memento of their surreal experience in the London Tube and exchange phone numbers. They will possibly meet again in a few days at a party. John is the only passenger left after all his new friends exit the carriage and go on to reach their final destinations.

Despite the optimistic end of the play, *Get Back* is a good example of how Galician arts are offering counter-discourses to challenge the over-positive view of migration shown by the media. It illustrates the opposition between these two discourses, that is, between a pleasant and a critical attitude towards the new Galician diaspora. Iria and Rafa's precariousness in their host country, and their home country before they decided to emigrate, can be seen to clash with their expectations for the future. They represent a generational clash: the failure of a generation that spent time and effort educating themselves, only to see their dreams fail to materialise. The critical attitude of the characters, who embody the

new Galician diaspora, questions the validity of the United Kingdom as a multicultural panacea. Brexit, indeed, increases their alienation in their host country, although racism is not exclusive of British citizens: John is British, and he is not a racist; Pepe is not British, and he is clearly a racist. This turn of the screw works efficiently to show the complexity of an ambivalent migrant subject such as Pepe.

Ameixeira relies on the Galician history of migration to write a play that offers irreconcilable views of migration. The idea of the 'good' and the 'bad' migrant conditions several stereotypes exposed in the previous analysis. On the one hand, male characters represent the 'good' migrant who does not complain about their situation in their host country, despite how precarious it might be as for Rafa. They are the ones who reject the 'something for nothing culture' that motivated part of the Brexit narrative. On the other hand, female characters have an open critical attitude towards their situation as subaltern commodified migrant subjects in the United Kingdom. Iria and Luisa illustrate the other side of the coin of migration—the unsuccessful and regretful one. Unfortunately, their voices are not heard in the media through programmes focused on the history of success. This omission is possibly why contemporary literature and documentaries, among other artistic manifestations, are giving a voice to these silent experiences of precariousness and frustration that are also part of the new migrant reality. The reality of a generation that does not accept the outcome forced on it by the different economic crises that marked their fate both in their home and host countries.

Thus, there is a generational and gender gap between the characters of the play. They only seem to have in common their country of origin and the space they share in the same carriage for a few hours during that sad day represented on stage. The London Tube represents this diaspora space that Ameixeiras needs to open a debate on expressing the new Galician diaspora through the arts. The different characters in the play are constructed through several stereotypes that encourage discussion on how new and old diasporas relate. Galician fractionated reality is accurately represented in a play that contributes to widening the image of contemporary migration through a play that is undoubtedly uncanny.

**Funding:** This research was funded by the Secretaría Xeral de Universidades, Xunta de Galicia, and it was part of the grant received by the project "Narratives of the New Diasporas".

**Acknowledgments:** Thanks to the author Diego Ameixeiras for sharing the unpublished text of the play for this analysis.

**Conflicts of Interest:** The author declares no conflict of interest.

## Notes

[1]  The line-up for the premiere of the play in Narón (on the north of A Coruña) was formed by Alfonso Agra, Federico Pérez, Tamara Canosa, Miguel Guido and Mercedes Castro.

[2]  Similarly, there is also a total clash between Pepe and Luisa, as well as between Rafa and Iria. As illustrated below, the play offers a dual gender and generational dichotomy.

[3]  Other examples of how the new Galician diaspora is being represented through the arts could be Eloy Domínguez Serén's (2015) biodocumentary *No Cow On The Ice*, the novel *Andrea contra pronóstico* by Alba Lago (2016), the song "*Emigrante*" by Pedro Caxade (2016), the short film *To Be and To Come Back* by Xacio Baño (2014) or the performance *Emigrantas* by Kirenia Martínez Acosta (2020), to name but some.

[4]  Although John is the only British-born character in the play, he speaks fluent Portuguese due to a relationship he had with a Brazilian girlfriend. Him speaking in Portuguese, instead of in English, allows the play to run smoothly without using English as the *lingua franca* of all the characters.

[5]  There is an obvious connection here between the old and new Galician diasporas and other migratiory experiences, globally speaking. From a general perspective, there are a number of 'pull' and 'push' factors that coincide with what Gordon K. Lewis (1990) points out as characteristics of certain migratory waves in the past and present. For instance, Lewis identifies poverty, unemployment and political instability as the main 'push' causes for migration for certain diasportic communities (the Caribbean one in his particular case) in the past, whereas the improvement in the means of transport and the wish to have access to a

Western education or the fulfilment of a better life are some of the 'pull' factors that he mentions when analysing contemporary diasporas.

6　For Gustavo San Román (2020, p. 478), 'morriña' implies "a profound feeling of lack [ ... ] sorrow about the absence, and metaphorically the dead, of a homeland". However, Helena Miguélez-Carballeira (2014) reinterprets the rhetorical use of what she calls 'Galician sentimentalism', implicit in 'morriña' as a disempowering tool that has historically marked Galician identity, also in the diaspora. These two views of 'morriña' although dissimilar, can complement each other when analysing the different characters in *Get Back*.

7　Icíar Bollaín's (2014) documentary *En tierra extraña* is a good indicator of the disappointment of this new generation of migrants. For further information about this documentary, see Alonso Alonso (2018).

8　Both Iria's and Rafa's generation are conditioned by the global economic crisis of 2007, which hit Spain in an unprecedented way. According to official data (INE n.d. and IGE n.d.) the number of Galicians living abroad had doubled since the beginning of this new century until just before the 2020 Covid pandemic, which implies that more than 300,000 people emigrated from Galicia in the last twenty years.

9　Obdulia Taboadela Álvarez (2007) already pointed out that the depreciation of Galician migrants as human capital was already in place during the second half of the twentieth century, when Galician professionals in countries such as Switzerland, Belgium or Holland, or cities such as Barcelona, Madrid or Bilbao underwent through the same complicating experience for the simple fact of being migrants.

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
