# Peer review of "Get Back: The New Galician Diaspora Goes on Stage"

_humanities, doi:10.3390/h10040111_

Round 1

Reviewer 1 Report

In this article the author(s) analyze Get Back (2016) a Galician play written by Diego Ameixeiras and directed by Jorge Coira. The article explores issues related to migration, especially generational and gender gaps, and the complexities of the concept of a New Galician diaspora while it situates the play within the frame of an early Brexit narrative where the London Tube is analyzed as a diasporic space. This is a very significant theme that presents a new dialogic dimension to the discursive analysis of the play.   

Overall, this is an original and well researched article (with multiple bibliographical references and all relevant). There is a strong introduction which identifies the topics of the paper as well as provides an essential context and the conclusion summarizes the key discursive topics discussed in the body of the article. In addition to being well structured and documented, the paper is well formulated and connected and it brings new and more comprehensive approaches to several fields of studies such as Galician diaspora, migration, the Brexit narrative, and the construction of “glocal” identities. This research will contribute in significant ways to new scholarly studies, and it will stimulate further debates among academic specialists and students. 

Author Response

Thanks so much for your comments. I have tried to expand the theoretical framework of the introductory part of the essay in order to engage with more critical material. Thus, I have added references to Vertovec’s ‘transnationalism’ and Appadurai’s ‘imaginary homelands’, together with note 5 which refers to Lewis’s ‘pull’ and ‘push’ factors of old and new diasporas. I have also expanded my approach to Brah’s concept of ‘diaspora space’ in order to connect it with my avenue of research on the new Galician diaspora.

Reviewer 2 Report

This article uses the play Get Back (2016) by Diego Ameixeiras to illustrate how the different characters embody aspects of what have been called the old and new Galician diaspora. It convincingly displays the complexities underlying these diasporic movements, as opposed to their simplified representation in mass media.  Therefore, I think this study is highly relevant, not only because it provides an original comparative approach to diaspora but also because it speaks directly to current issues affecting British and European social relations.

I believe the content is well structured, all the information is clearly presented (research questions as well as results) and the literary analysis is sharp and meaningful. Here are a few points that I think could be improved in this article:

  1. One thing I miss is more engagement with critical material. Although I find the discussion about the old and new diasporas compelling and indeed one of the strongest points of the article, I was wondering whether it would be possible to back up the arguments with scholarly sources in more depth. For instance, I think Avtar Brah’s idea of the “diaspora space” remains underexplained (line 100). Also, the author says that “But Galician migrant experience has changed radically in the last decades. High-speed travel, digital technologies and a different historical context characterise the new Galician diaspora” (lines 118-120). I think this is not only the case of the Galician diaspora and this statement could indeed be applied to many other migrant movements today. Could the author connect this phenomenon to a more general context? Perhaps alluding to theorists who have dealt with how these changes have affected the way we talk about diaspora and migration (I’m thinking, for example, Steven Vertovec in Transnationalism as early as 2009). Also, when addressing the character of Pepe, who seems to embody the old Galician diaspora (lines 139-141), I think a concept such as Salman Rushdie’s “imaginary homelands” could be usefully applied; or his experiences could be best articulated through some of the earliest formulations of diaspora (William Safran’s or Khachig Tölölyan’s) to reinforce this idea of the “old” diaspora.
  2. In a couple of places I think I’ve spotted some out-of-place Galician: in the first footnote “The line-up for the premiere of the play in Narón was formed by Alfonso Agra, Federico Pérez, Tamara Canosa, Miguel Guido e Mercedes Castro” I believe that “e” should be replaced by “and”; similarly, in line 44 there is an “ou” that I think should be “or”.

Apart from these minor adjustments I think the article is definitely worth publishing so, in my opinion, it will be ready for publication after a more in-depth engagement with diaspora scholarship.

Author Response

(The authors gave the same response as above.)
